# Triterpenoids in Jujube: A Review of Composition, Content Diversity, Pharmacological Effects, Synthetic Pathway, and Variation during Domestication

**DOI:** 10.3390/plants12071501

**Published:** 2023-03-29

**Authors:** Fuxu Pan, Xuan Zhao, Fawei Liu, Zhi Luo, Shuangjiang Chen, Zhiguo Liu, Zhihui Zhao, Mengjun Liu, Lili Wang

**Affiliations:** 1College of Horticulture, Hebei Agricultural University, Baoding 071000, China; 2Research Center of Chinese Jujube, Hebei Agricultural University, Baoding 071000, China

**Keywords:** jujube, sour jujube, triterpenoids, types, pharmacological activities, synthetic pathway, domestication

## Abstract

Chinese jujube (*Ziziphus jujuba* Mill.) and its wild ancestor, sour jujube (*Z. acidojujuba* C.Y. Cheng & M.J. Liu), is a Ziziphus genus in the Rhamnaceae family. ZJ and ZA are rich in a variety of active ingredients, with triterpenoids being a unique active ingredient, which are present in the fruit, leaves, branches, and roots. More than 120 triterpenoids have been identified in ZJ and ZA, and have various biological activities. For example, betulinic and ursolic acids have anticancer, antioxidant, antibacterial and antiviral activities. ceanothic, alphitolic, and zizyberanalic acids possess anti-inflammatory activities. The MVA pathway is a synthetic pathway for triterpenoids in ZJ and ZA, and 23 genes of the MVA pathway are known to regulate triterpene synthesis in ZJ and ZA. In order to better understand the basic situation of triterpenoids in ZJ and ZA, this paper reviews the types, content dynamic changes, activities, pharmacokinetics, triterpenoid synthesis pathways, and the effects of domestication on triterpenoids in ZJ and ZA, and provides some ideas for the future research of triterpenoids in ZJ and ZA. In addition, there are many types of ZJ and ZA triterpenoids, and most of the studies on their activities are on lupane- and ursane-type triterpenes, while the activities of the ceanothane-type and saponin are less studied and need additional research.

## 1. Introduction

Chinese Jujube (*Ziziphus jujuba* Mill.) is domesticated from its wild ancestor sour jujube (*Z. acidojujuba* C.Y.Cheng & M.J.Liu). A study found that the genome of sour jujube (ZA) is larger than that of jujube (ZJ), which is a variation that occurs during the domestication process, resulting in a considerable difference in the genome size between ZJ and ZA [1]. ZJ and ZA have a high nutritional value and are rich in various active ingredients, including vitamin C, phenols, flavonoids, polysaccharides, triterpenes, saponins, vitamins, and other active ingredients [2,3,4]. Chinese jujube is native to China, and it is the most important member of the Rhamnaceae family in terms of its growing area and high value as a fruit tree and medicinal plant [5]. According to two famous ancient medicine books, “Shennong’s Herbal Classic” and “Compendium of Materia Medica”, jujube fruit nourishes the spleen and stomach, replenishes qi and blood, soothes nerves, provides medicinal properties, and prolongs life [6].

Triterpenoids are one of the most characteristic and major active components of the Ziziphus genus, mainly in the form of triterpenes and saponins [7]. Several review articles have been published describing the triterpenoids of the Ziziphus genus. These articles mainly describe the multiple active components of the Ziziphus genus, and triterpenoids, as a segment of them, usually provide only a brief overview of their species and activities [8,9,10,11]. Additionally, this paper focuses on the triterpenoids of ZJ and ZA, which not only complement the species and activity of triterpenoids of the Ziziphus genus, but also describe in detail the dynamic changes in content, pharmacokinetics, triterpene synthesis pathways, and the effect of domestication on triterpenoids of ZJ and ZA. This is the first study to conduct such an analysis. The review of this paper can provide some references for the research trends of ZJ and ZA triterpenoids and expand our understanding of Ziziphus genus triterpenoids.

In addition, triterpenoids, as an important active ingredient in plants, have high lipid solubility, with molecular weights generally in the range of 400–600. The side-chain structure of triterpenoids changes considerably, such as hydroxylation, epoxidation, cyclization, carbon reduction, and the formation of double bonds. The change in side chains leads to the diversity of triterpenoids’ structure [12]. Due to the structural diversity of triterpenoids, they present strong pharmacological activities, such as anti-cancer, anti-inflammatory, and anti-diabetic activities [13,14], which makes the triterpenoids of ZJ and ZA potential targets for the development of new drugs.

## 2. Results

### 2.1. Types of Triterpenoids in Jujube

Triterpenoids are widely distributed, primarily, in a free state or in the form of esters, ethers, and glycosides, which have high lipid solubility. These triterpenoids are generally terpenoids consisting of 30 carbon atoms, and the isoprene rule states that most triterpenes are the condensation of 6 isoprenes (30 carbons). According to the carbocyclic structure, these triterpenoids may be divided into several types, such as chain, monocyclic, bicyclic, tricyclic, tetracyclic, and pentacyclic triterpenes [15]. Triterpenes may be divided into free triterpenes and triterpene saponins, based on their ability to bind to sugars [16].

#### 2.1.1. Triterpenes

The free triterpenes in ZJ and ZA are primarily pentacyclic triterpenoids, including lupane-, oleanane-, ursane-, and ceanothane-types. The chemical structures of various types of triterpenes are presented in Figure 1.

There are five- and six-membered carbocyclic rings in the structure of lupane-type triterpenoids. The E ring is a five-membered carbocyclic ring, while the A, B, C, and D rings form the skeleton structure of the six-membered carbocyclic ring. The 19-position isopropyl of the E ring is substituted by an α configuration. According to its structure type, lupane-type triterpenoids are divided into basic structure, ring-opening, lactone-forming, and other categories [17]. Guo [18,19], Fujiwara [20], Lee [21], He [22], and Wang [23] analyzed the chemical structure of lupane-type triterpenoids, including betulinic acid, betulin, and lupeol, isolated from the fruits and leaves of ZJ and ZA using nuclear magnetic resonance (^1^H-NMR, ^13^C-NMR), tandem mass spectrometry (MS), electrospray ionization (ESI), and electrospray ionization mass spectrometry (ESI-MS).

Oleanane-type triterpenes, also known as β-amyrin-type triterpenes, are the most common and widespread type of triterpenoid skeleton in the plant kingdom. They generally exist in the form of aglycones or glycosides. The five rings in their triterpene structure are generally six-membered rings. Guo [18] identified oleanolic, maslinic, and hydroxyl oleanolic acids in ZA fruit by analyzing the ^1^H, ^13^C-NMR and ESI-MS data and comparing them with the previously reported data. Lee [21] and Guo [24] found five oleanane-type triterpenes in ZJ.

Ursane-type triterpenes are also called α-aromatic resin alkane-type triterpenes. The five rings in the structure are generally six-membered rings, which are similar to oleanane-type triterpenes. The main difference between ursane- and oleanane-type triterpenes is that the two methyl substituents are present at the C-19 and C-20 positions, respectively. To date, ursane-type triterpenes obtained from ZJ and ZA are pomoldic acid, ursolic acid, 2α-hydroxyursolic acid/corosolic acid, cecropiacic acid, 2α, 3β, 13β, 23-tetrahydroxy-ursul-11-ene-28-carboxylicacid,2α,3β-dihydroxy-urs-20(30)-en-28-oic acid, 2α, 3β, 28-trihydroxy-urs-20(30)-ene, and 3β, 12β, 13β-trihydroxy-ursan-28-oic acid [25,26].

Ceanothane-type triterpenes are relatively rare in plants, and most are found in the Rhamnaceae family [27,28]. A total of 27 ceanothane-type triterpenes have been reported ZJ and ZA, and ceanothic acid is the most common ceanothane-type triterpene.

According to the published literature, 79 pentacyclic triterpenoids were isolated and identified in ZJ and ZA, of which ceanothane-type triterpenes were the most abundant, with 27 types. Oleanane-type triterpenes were the least abundant, with 10 types. Lupane-type triterpenes had 22 types and ursane-type triterpenes had 20 types. The specific distributions are shown in Table 1.

#### 2.1.2. Saponins

Saponins are a class of structurally complex, natural products formed by the condensation of glycosomes (one or more sugar chains) and glycogens (triterpenoids, steroids, or steroidal alkaloids). The triterpene glycogens of ZJ and ZA are mostly dammarane-type tetracyclic triterpenes, and their glycoside parent nuclei may be divided into various types, according to whether the side chain is looped, as well as the position of double bonds. Glycans are mostly substituted in the C-3, C-20, and C-23 positions and are generally linked to glycosyls, such as rhamnose, arabinose, glucose, galactose, glucose, glucose, and other glycosophyls. Triterpene saponins are divided into types I, II, III, IV, and others, according to the structure of the parent nucleus [37]. Type I triterpenoid saponins are the most isolated and commonly identified in ZJ and ZA, with a total of 23 species. Only 1 species for types III, VII, VIII, IX and XI was isolated and identified. The chemical structure types and types of triterpene saponins of ZJ and ZA are shown in Figure 2 and Table 2.

### 2.2. Dynamic Changes in Triterpenoid Content in Jujube and Sour Jujube

The triterpene contents of ZJ and ZA varied considerably in different regions and for different varieties. The hierarchical clustering analysis (HCA) of 10 types of triterpene acids in 42 ZJ fruit samples from 22 different planting areas revealed that most of the contents of ZJ triterpenoids in the same planting area did not differ considerably, and the triterpene content of ZJ in different planting areas varied greatly, which indicated that the variation in the triterpene content of ZJ was related to the cultivation area [54]. The UPLC-MS method was used to determine the content of five main triterpene acids in 99 ZJ varieties in the germplasm resource base of Tarim University, and the results showed that the content of five triterpene acids in 99 ZJ varieties varied in range. Betulinic acid was 516.409~4097.962 µg/g, alphitolic acid was 198.195~3282.203 µg/g, maslinic acid was 13.905~751.855 µg/g, oleanolic acid was 36.696~837.463 µg/g, and ursolic acid 5.267~685.325 µg/g [55]. Oleanolic or ursolic acids were not detected in a small number of varieties of ZJ fruit, such as ‘Hebeicuizao’, ‘Beijingeggzao’, and ‘Binxianblackpimplezao’ [56]. The total triterpene content of 50 ZJ cultivars in the same area was measured spectrophotometrically, and the results showed that the total triterpene content of 50 ZJ cultivars was between 3.99~15.73 mg/g [57]. The content of ZA triterpenoids also varied in different regions, and the total triterpene and three main triterpene acids of 16 ZAs in Shanxi, Shaanxi, and Liaoning were determined. The total triterpene content of ZA was 1.93~3.96 mg/g, oleanolic acid was 180~920 µg/g, ursolic acid was 180~540 µg/g, and betulic acid was 270~1330 µg/g [58].

The triterpene content in different tissue parts of ZJ and ZA varied. The triterpene content in ZJ leaves was much higher than the other parts of ZJ trees, and the total triterpene content of ZJ leaves was the highest at 41,570 µg/g, followed by ZJ fruit. The total triterpenoid content of ZJ flowers was the lowest at 8.499 µg/g, and the total triterpenoid content between ZJ leaves, fruits, and flowers was significantly different [59]. In the determination of total triterpenoids in ZJ flowers, buds, young leaves, mature leaves, and annual stems of ZA and ZJ, the total triterpenoid content in the buds was the highest, with 14.979 mg/g for ZA and 13.080 mg/g for ZJ, and the order of total triterpene content in ZA and ZJ was bud > stem > young leaves > mature leaves > flowers [60].

The content of various active substances in ZJ fruit at different growth stages was affected by the growth and development of the fruit. The fruit development periods of ZA and ZJ were divided into six stages: young fruit, expansion stage, white ripening stage, first-red, semi-red, and full-red ripening. The dynamic accumulation of total triterpene content during development was measured. The results show that the triterpenoids of ZA and ZJ first increase and then decrease, and the total accumulation of triterpenoids in ZA and ZJ fruits peaks during the white ripening stage, with contents of 16.057 and 8.795 mg/g, respectively, which increases 5.3 and 2.9 times compared to the triterpene content in the young fruit stage [60]. Another study divided the ZJ fruits of ‘Jinsixiaozao’ into six growth and maturity periods, from S1 to S6, according to color and size, and the total triterpenoids in the S1 to S3 periods showed an upward trend. The total triterpenoid content in the S3 period peaked, and the triterpene content in the S4 to S6 fruits gradually decreased [61]. The triterpenoid content in ‘Goutouzao’ was the highest when the fruit was semi-red period, and the triterpene content gradually decreased as the fruit’s maturity increased [62]. The total triterpene content in ‘Wuyizao’ was the highest when it was in the white ripening stage, the total triterpene content in the fruit was significantly reduced during the crisp ripening stage, and the total triterpene content increased slightly, compared to the brittle ripening stage, when the ‘Wuyizao’ fruit transformed into the full-red stage [63]. The total triterpene content in green ZA fruits was higher than that of ripe ZA fruits [64]. These results show that the increase in the diameter of jujube fruit during its growth has a minor effect on the fruit’s triterpenoid content. With the continuous development and growth of the fruit, its triterpene content continues to increase, and then gradually decreases when it reaches a maximum level [65,66].

### 2.3. Biological Activity of Triterpenes in Jujube and Sour Jujube

#### 2.3.1. Anti-Cancer Activity

Triterpenoids (3-O-trans-p-coumaroyl alphitolic acid) in ZJ fruit reduce the viability of human lung adenocarcinoma cells (A549), prostate carcinoma cells (PC-3), and human breast cancer cells (MDA-MB-231) in a concentration-dependent manner. The IC50 values range from approximately 9 to 12 μM. 3-O-Trans-p-coumaroyl alphitolic acid induces the apoptosis of these cancer cells by increasing mitochondrial ROS production [67].Yang [68] extracted triterpene components from ‘Jinsixiaozao’ and observed that they significantly inhibited the proliferation of prostate cancer cells, and this inhibitory effect was concentration- and time-dependent. Eighteen active components were extracted from ZJ fruits and leaves. Among them, pomlic, 3-O-trans-p-coumaroyl alphitolic, 3-O-cis-p-coumaroyl alphitolic, and betulinic acids had strong inhibitory effects on breast cancer (MCF-7), lung adenocarcinoma A549, human liver cancer (HepG2), and colon cancer (HT-29) cells, presenting the highest inhibition rate of 99%. However, oleanolic acid only inhibited breast cancer (MCF-7) and lung adenocarcinoma (A549) cells with an inhibition rate higher than 90%; however, it had no inhibitory effect on human hepatoma (HepG2) or colon cancer (HT-29) cells [69]. Qiao [25] isolated 12 triterpenoids from ZA and used the MTT method to determine the anti-proliferative activity of triterpenoids in ZJ on cancer cells. Triterpenoids 7 (2α, 3β, 19α-trihydroxy-urs-12-en-28-oic acid, 2α), 9 (3β-dihydroxy-urs-20(30)-en-28-oic acid) and 10 (2α, 3β, 28-trihydroxy-urs-20(30)-ene) presented strong inhibitory activities on the proliferation of human hepatoma (HepG2) and human breast cancer (MCF-7, MDA-MB-231) cells with IC50 values less than 10 μM. Compounds **11** (3β, 12β, 13β-trihydroxy-ursan-28-oic acid) and 12 (2α, 3β, 12β, 13β-tetrahydroxy-ursan-28-oic acid) presented significantly reduced anti-proliferative activities due to the oxidation of the double bond on the C ring and two hydroxyl substitutions. The solubility of betulinic acid (BA) extracted from ZA was low, and the co-precipitation method was combined with β-cyclodextrin (β-CD), which enhanced the solubility of BA. BA-b-CD had an obvious anti-proliferative effect on MCF-7 (breast adenocarcinoma) cells. BA-b-CD induced Bax expression and inhibited Bcl-2 expression to decompose the mitochondrial outer membrane, release cytochrome C, and activate the downstream apoptosis cascade, which inhibited the proliferation of breast adenocarcinoma cells [70].

#### 2.3.2. Antioxidant and Hepatoprotective Activities

When ROS (reactive oxygen species) produced in the body exceeds the scope of the antioxidant protection system, it damages the body, and numerous diseases are related to oxidative stress [71]. The polysaccharides, flavonoids, and triterpenoids of ZJ and ZA have scavenging effects on DPPH free radicals, ABTS free radicals, and hydroxyl free radicals, and exert certain antioxidant activities [72,73].

The liver is an important metabolic organ in the body, and it participates in various roles. Liver damage is the main cause of liver disease. Continuous liver damage leads to liver disease and functional weakness, liver fibrosis and cirrhosis, and liver cancer in severe cases [74]. Total triterpenes (TTs) and BA in red ZJ presented protective effects on the establishment of a mouse alcoholic liver injury model for 30% ethanol gavage prepared from liquor. Their results show that ROS are one of the causes of alcohol-induced liver damage, and BA and TTs inhibit the accumulation of lipids in the liver, prevent and repair liver cell membrane damage caused by alcohol, and maintain the liver antioxidant system to enhance the removal of harmful free radicals caused by ROS, which protects the liver [75]. Cecal ligation and perforation (CLP) causes liver failure and damage. Jujuboside B in ZJ regulates and reduces the inflammatory response caused by CLP, increases the expression of the glucocorticoid receptor (GR), improves the endogenous antioxidant capacity, and protects against liver damage [76].

Triterpenoids were extracted from red ZJ and ZA, and the scavenging rates of DPPH · and H_2_O_2_ were analyzed. It was found that the triterpenoids had a certain scavenging effect on free radicals and presented good antioxidant activity [77]. Cai et al. [78] reported that the ability of purified ZJ triterpenes to scavenge ABTS+ and OH was higher than crude extracts; however, the ability to scavenge DPPH was reduced compared to the crude extract. Sun et al. [79] extracted and purified triterpenoid saponins from ZA fruit, and their results show that triterpenoid saponins have a significant ability to scavenge DPPH free radicals, inhibit the formation of the lipid peroxidation product MDA, and protect biological membranes from lipid peroxidation damage. Qiao [25] et al. used the peroxyl radical scavenging capacity (PSC) assay and the superoxide anion radical (O^2−^) assay to detect the antioxidant activity of 12 triterpenoids in ZA. Compound **10** (2α, 3β, 28-trihydroxy-urs-20(30)-ene) presented the highest antioxidant activity in the PSC assay with an EC50 of 0.8 ± 0.02 μM, and its antioxidant activity was 18.9 times greater than of ascorbic acid. Compound **9** (2α,3β-dihydroxy-urs-20(30)-en-28-oic acid) showed the highest free radical scavenging activity in the superoxide anion radical (O^2−^) assay with an EC50 of 6.1 ± 0.05 μM. The hydroxyl group (carboxyl group) on C-28 in ZJ triterpenoids was crucial to the antioxidant activity of ZJ triterpenoids.

#### 2.3.3. Anti-Viral Activity

Hong [80] established an in vitro model of ZJ betulinic acid against influenza A/PR/8 virus and found that the betulinic acid inhibited the proliferation of the influenza A/PR/8 virus. The administration of drugs to mice infected with the A/PR/8 virus revealed that ZJ betulinic acid significantly reduced the inflammation and pulmonary oedema caused by it. The influenza virus relies on the hemagglutinin protein (HA) on the surface of the virus to recognize and bind to sialic acid in the host’s respiratory tract, which results in the infection of the host by the virus. However, betulinic acid-derived compounds bind with the hemagglutinin protein (HA) to inhibit the activity of the influenza virus [81,82].

#### 2.3.4. Immune and Anti-Inflammatory Activities

Zhang [83] performed immunological experiments on the polysaccharides, polyphenols, and triterpenoids of Xinjiang ‘Ashoke’ ZJ, and found that the triterpenoids promoted lymphocyte proliferation and significantly promoted LPS and IFN-γ co-stimulated macrophage proliferation. The triterpenoids of ‘Ashoke’ ZJ enhanced the cellular, humoral, and non-specific immunity factors of mice, and significantly improved various immune indicators in the immune experiments on mice, such as the immune organ index and phagocytosis index α [84].

Lipopolysaccharides (LPSs) stimulate cells to release NO in the form of nitrite (NO^2−^) in the medium, and the production of nitrite leads to an endotoxin-induced inflammatory response in cells. Masullo et al. [85] measured the effects of four triterpenoids, including ceanothic, alphitolic, 3-O-trans-p-coumaroyl alphitolic, and oleanolic acids, on the release of nitrite, which is the stable end-product of NO production in lipopolysaccharide-activated mouse macrophages. Their results showed that two triterpenes, alphitolic and oleanolic acids, had a significant inhibitory effect on NO production, with the highest inhibition rate reaching over 95%. Alphitolic and oleanolic acids inhibit the expression of nitric oxide synthase (iNOS), which reduces the endotoxin inflammatory response. The inhibitory effect of triterpenes extracted from ZJ on nitric oxide (NO) was related to the triterpene skeleton, and oleanolic triterpenoids and ursulane-type triterpenes had a good effect on NO inhibition. 2-O-Trans-p-coumaroyl maslinic and 3-oxo-urs-12-en-28-oic acids had no inhibitory effect on RAW 246.7 cell activity at 5 µM and inhibited the production of NO, both of which were passed. The NF-κB signaling pathway exhibited anti-inflammatory activity by downregulating the expression of inflammatory factors [33]. Another study reported that triterpenoid saponins moderately inhibited the release of the proinflammatory cytokine TNF-αin LPS-induced RAW 246.7 macrophages [48]. *Euphorbia fischeriana* extracted stimulated cells to present inflammatory responses, and the triterpenoids in ZJ fruit had a significant inhibitory effect on the inflammatory cells activated by *Euphorbia* extract. Zizyberanalic acid presented the best effect on the inhibition of the pro-inflammatory cytokine TNF-α and exhibited no cytotoxicity [86].

#### 2.3.5. Other Biological Activity

Triterpenes from ZJ and ZA also presented other pharmacological activities. Sun [79] isolated and purified saponin compounds from ZA and performed anti-bacterial activity tests on eight strains, including *Staphylococcus aureus*, *Bacillus subtilis*, and *Streptococcus pneumoniae*. The saponin compounds from ZA had obvious inhibitory effects on the eight strains of pathogens and broad-spectrum antibacterial and antifungal effects. Kawabata et al. [87] found that betulinic acid extracted from ZJ effectively promoted glucose uptake by skeletal muscle cells. Lee et al. [88] isolated a new triterpene, 3-dehydroxyceanothetric acid 2-methyl ester (3DC2ME), from the ZJ root, and 3DC2ME prevented cisplatin-induced damage to LLC-PK1 nephrocytes and protected LLC-PK1 nephrocytes from Cisplatin-induced renal cytotoxicity by modulating the MAPK and apoptotic pathways. Fujiwara et al. [20] showed that triterpenoids in ZJ inhibited the formation of foam cells in macrophages and reduced the occurrence of cardiovascular diseases. ZJ saponin B (JB) extracted from ZA fruit effectively reduced the number of inflammatory cells in the bronchoalveolar lavage (BAL) fluid, reduced the severity of lung inflammation, and had a therapeutic effect on oval protein (OVA)-induced allergic asthma in mice, and JB has a potential role in the treatment of asthma [89]. The Biological Activities are summarized in Table 3.

### 2.4. Pharmacokinetic Study of Jujube and Sour Jujube Triterpenoids

Pharmacokinetics is primarily used to study the differences in the pharmacokinetic parameters of the absorption, distribution, metabolism, and excretion of drugs into the body between the pathological and normal states, which provide a basis for understanding the dynamic changes in drugs in the body and the effective guidance on rational clinical drug use [90,91].

Li et al. [92] used ultra-performance liquid chromatography-mass spectrometry (UPLC-MS/MS) to determine the changes in seven triterpene acids in the plasma of normal and immunosuppressed rats after the oral administration of jujube total triterpene extracts (TAEs). The average plasma concentration–time curves of the seven triterpenoids in normal and immunosuppressed rats were similar. However, the pharmacokinetic parameters of triterpenoids in normal and immunosuppressed rats were different, especially the peak concentration (Cmax), which exhibited significantly lower levels in immunosuppressed rats. The area under the plasma concentration–time curve (AUC0−t and AUC0~∞) of epiceanothic acid obtained from the drug concentration–time curve was significantly higher in the immunosuppressed rats after the oral administration of TAEs than in normal rats. The maximum blood concentration (Cmax) and apparent plasma clearance (CLz/F) of epiceanothic acid significantly decreased in immunosuppressed rats. These results indicate that immunosuppression can increase the bioavailability of epiceanothic acid and decrease its elimination rate. In contrast, the mean AUC0−t, AUC0~∞, and mean peak concentration (Cmax) of alphitolic acid in immunosuppressed rats showed relatively lower values compared to those in normal rats. The CLz/F of alphitolic acid was markedly higher in immunosuppressed rats than in normal rats. These results suggest that systemic exposure (AUC and Cmax) decreases and its elimination increases in immunosuppressive states. In addition, other pharmacokinetic parameters were found to be different; however, this was not significant compared to those in the normal rats.

The same research group also revealed the dynamic changes in triterpene acids in the state of pathological liver injury in normal and CCl4 acute liver injury rats. The results show that there are significant differences in the pharmacokinetic parameters of the seven triterpenoid acids between the liver injury model and normal groups. In the pathological state of acute liver injury, the area under the time curve (AUC0~t and AUC0~∞) for epicaschuic and pomeloic acids was significantly higher than that in normal rats, and their drug clearance rates (CLz/F) were significantly reduced, indicating that acute liver injury can improve the utilization rate of these two triterpenoid acids and reduce their elimination ability. AUC0~t, AUC0~∞ and the mean peak concentrations (Cmax) of alphitolic, betulinic, and betulinic acids in acute liver injury were significantly lower than those in normal rats, while CLz/F was significantly increased. These results indicate that the utilization rate of these three triterpenoid acids is reduced and the elimination ability is enhanced [93].

Du et al. [94] conducted pharmacokinetic studies on the water extract of ZJ seeds containing six compounds, including jujube saponins A (JuA) and B (JuB) in normal and insomniac rats. The study found that JuA and JuB were more rapidly absorbed in the insomnia model (IM) mice than in normal control (NC) rats, and the absorption rate and degree for the insomnia model mice increased, compared to the normal control group, Cmax and AUC increased and Tmax was significantly reduced, which was beneficial to the therapeutic effect for insomnia. While the plasma clearance (CL) value of JuB was much higher than that of JuA, it may be related to the hydrolysis of saponins mediated by bacteria in the gastrointestinal tract after oral administration, because JuA is hydrolyzed to JuB in the intestinal segment, and then further metabolized to ZJ saponins by mouse intestinal bacteria, indicating that the hydrolysate may be a biologically active absorption form [95].

### 2.5. Synthetic Pathways and Regulation of Jujube and Sour Jujube Terpenoids

Triterpenoids are widely present in plants, and there are two synthesis pathways for triterpenoids in plants: MVA and MEP [96]. The triterpene synthesis pathways of ZJ and ZA are the MVA synthesis pathway. The MVA synthesis pathway first consists of acetyl-CoA as the starting substrate, and after six reaction stages to generate isopentenyl pyrophosphate (IPP), IPP, and dimethylallyl pyrophosphate (DMAPP), are converted to farnesyl under the action of farnesyl pyrophosphate synthase (FPS). Diphosphate (FPP) was synthesized squalene catalyzed by squalene synthase (SQS), and squalene was catalyzed by squalene epoxidase (SE) catalysis to 2,3-oxidosqualene [97]. Various terpenoids are synthesized by the cyclisation, hydroxylation, and glycosylation of 2,3-oxidesqualene, and the enzymes that mediate these reactions include oxidosqualene cyclase (OSC), cytochrome P450 (CYP450), and glycosyltransferases (UGT) [98,99,100]. Based on the transcriptome data analysis of ZJ and ZA, 23 genes involved in the synthesis of ZJ and ZA triterpenoids were obtained, and the specific gene information and synthesis pathways are presented in the Figure 3 [60].

Plant triterpenoids are regulated by a variety of TFs, including WRKY TFs, bHLH TFs, AP2/ERF TFs, and bZIP TFs, regulated at the transcriptional level [101]. In addition, plant triterpenoids are induced by biological factors, such as methyl jasmonate (MeJA) and salicylic acid (SA) hormones, which induce terpenoid synthesis and promote the accumulation of triterpenoids [102,103]. Genome-wide analysis identified 61 ZJWRKY transcription factors in ZJ, and 19 ZJWRKY transcription factors were differently expressed in ZJ and ZA [104]. The accumulation of triterpenoids in ZJ was induced by MeJA and SA, and the expression of ZjWRKY18 was consistent with the accumulation of triterpenes induced by MeJA and SA, which indicated that ZjWRKY18 played an important role in the accumulation of triterpenes. The expression level of ZjWRKY18 was the highest in the roots, followed by the leaves and fruits, and it was the lowest in the stems. The silencing of ZjWRKY18 reduced the expression and accumulation of triterpenoids in ZJ. The overexpression of ZjWRKY18 promoted the synthesis of triterpenoids and upregulated HMGR, FPS, and SQS genes, which indicated that ZjWRKY18 regulated the synthesis pathway of triterpenoids in ZJ [105].

### 2.6. Effects of Domestication on Jujube and Sour Jujube Triterpene Metabolism

ZJ is domesticated from wild, sour jujube. ZA has sourer, smaller, and less fleshy fruit, and it gradually evolved into ZJ with a sweet taste, large fruit, and more flesh after a long period of artificial or natural domestication [106]. Metabolomic analysis of 182 ZJ and 25 wild ZA germplasm resources identified 427 metabolites with chemical annotations, including 41 terpene metabolites, and 90 metabolites showed a decreasing pattern from wild ZA to ZJ, and the reduced metabolites were primarily rich in triterpenoids. A comparison of contents of 22 triterpenoids showed that 15 were directly selected during the domestication process, especially undecane and ursulane types, and were significantly reduced from wild ZA to ZJ. The relevant sites of genomes and metabolites revealed a decrease in the frequency of alleles representing higher triterpene contents in wild ZA and ZJ, identifying a homologous cluster containing seven 2,3-oxidosqualene cyclases (OSCs), and found that OSC genes contributed to the reduction in triterpene content during domestication, which indicated that the negative selection of triterpenoids was the basis for ZJ domestication [107]. The results show that the total triterpene content of wild ZA is higher than ZJ [24,60], which demonstrates that domestication affects the expression of terpene genes and reduces the synthesis of triterpene metabolites in ZJ.

## 3. Conclusions and Perspectives

Triterpenoids are one of the main active components in ZJ and ZA and produce a variety of active effects. Therefore, they have received extensive attention from researchers. Prior to 2010, most of the literature focused on the isolation and structural identification of triterpenoids in ZJ and ZA. Since 2010, increased attention has been paid to the biological activity of triterpenoids in ZJ and ZA. According to the literature, there are more than 120 types of triterpenoids in ZJ and ZA, including 79 types of triterpenoids and 42 types of saponins. Jujube and ZA triterpene compounds have liver-protective, antioxidant, anti-tumor, anti-inflammatory, and other active effects. They are some of the most important lead compounds for the further development of novel drugs. There are many types of triterpenoids in ZJ and ZA; only several common triterpenoids have been studied for their biological activity, and most of the triterpenoids have not been reported for their biological activity. Further research on the activities of various triterpenoids in ZJ and ZA is needed, which can help us to discover and screen the triterpenoids with a good-activity effect and facilitate the development and application of innovative drugs for diseases. The low solubility and selectivity, poor bioavailability, and short half-life of triterpenoids require structural modifications to enhance their biological activity. The potentially low bioactivity levels of ZJ and ZA triterpenoids can be improved by structural modifications to enhance their active effects and increase the range of bioactivity.

The metabolism and accumulation of triterpenoids in ZJ and ZA are affected by production and cultivation, biosynthesis, and environmental factors. Studying these factors can regulate the synthesis of triterpenoids in ZJ and ZA, so as to better understand the synthesis pathway and mechanism of triterpenoids in ZJ and ZA. Due to the wide variety of terpenoids and their complex metabolic properties, it is necessary to further explore the genes related to ZJ and ZA synthesis and to determine the genes key to increasing the triterpenoid content. Furthermore, we must clarify the regulation pathway for high triterpenoid contents in ZJ and ZA, and provide an innovative theoretical basis and guidance for its production, so as to cultivate jujube and ZA varieties with a high triterpenoid content.

## 4. Materials and Methods

Various search engines and online databases (such as PubMed, CNKI, ScienceDirect, Sci Finder, and Web of Science) were used to search the scientific literature on jujube and ZA triterpenes, published from November 1991 to September 2022. The search terms used were jujube, sour jujube, triterpenes, saponins, biological activity, pharmacokinetics, growth stage, variety, and origin. The search languages were primarily English and Chinese. The types of articles retrieved by this study included research articles, reviews, Master’s and doctoral dissertations, including more than 80 research-based articles of various types, 14 review-based articles, and 5 Master’s theses.

## Figures and Tables

**Figure 1 plants-12-01501-f001:**
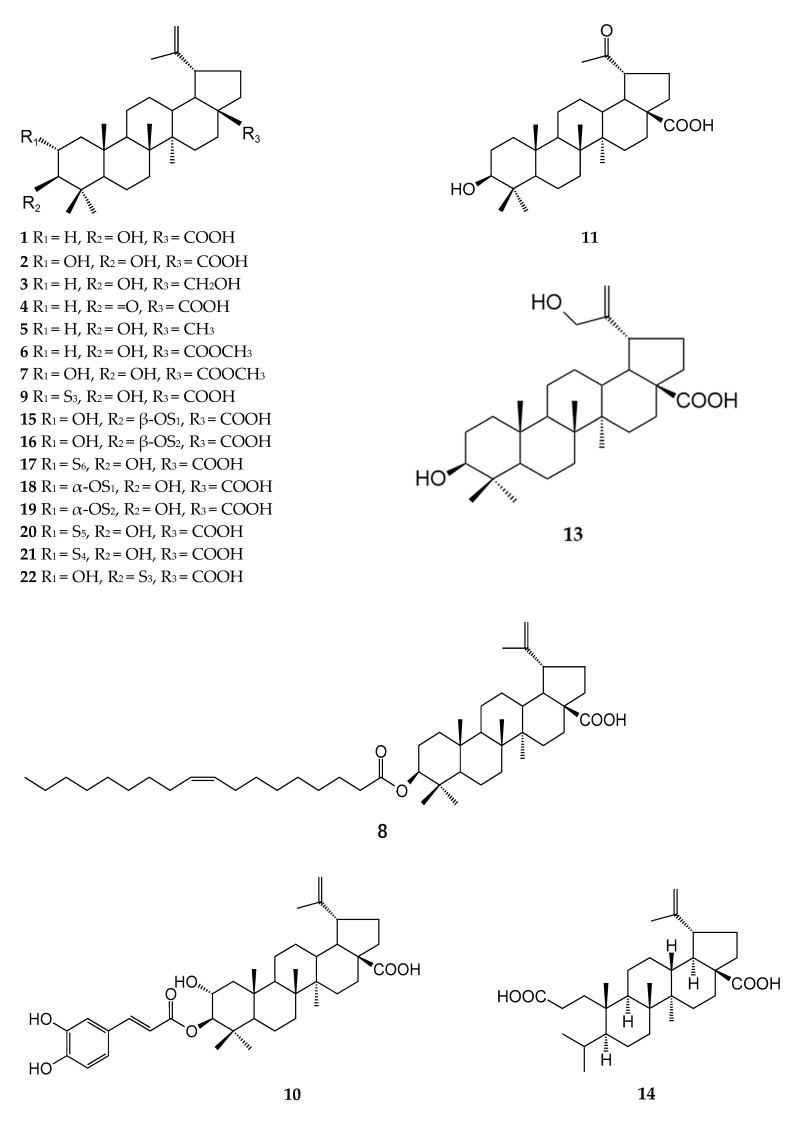
Chemical structures of triterpenoids in ZJ and ZA.

**Figure 2 plants-12-01501-f002:**
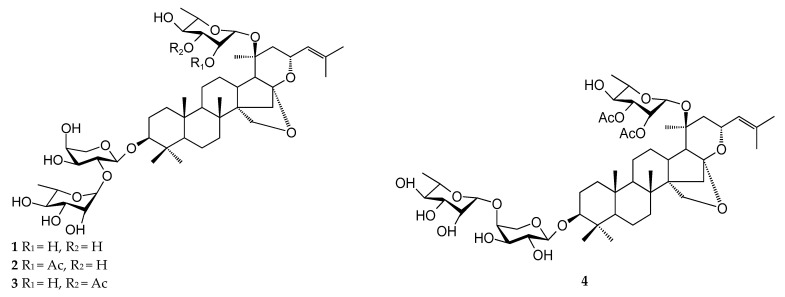
Chemical structures of triterpenoid saponins in ZJ and ZA.

**Figure 3 plants-12-01501-f003:**
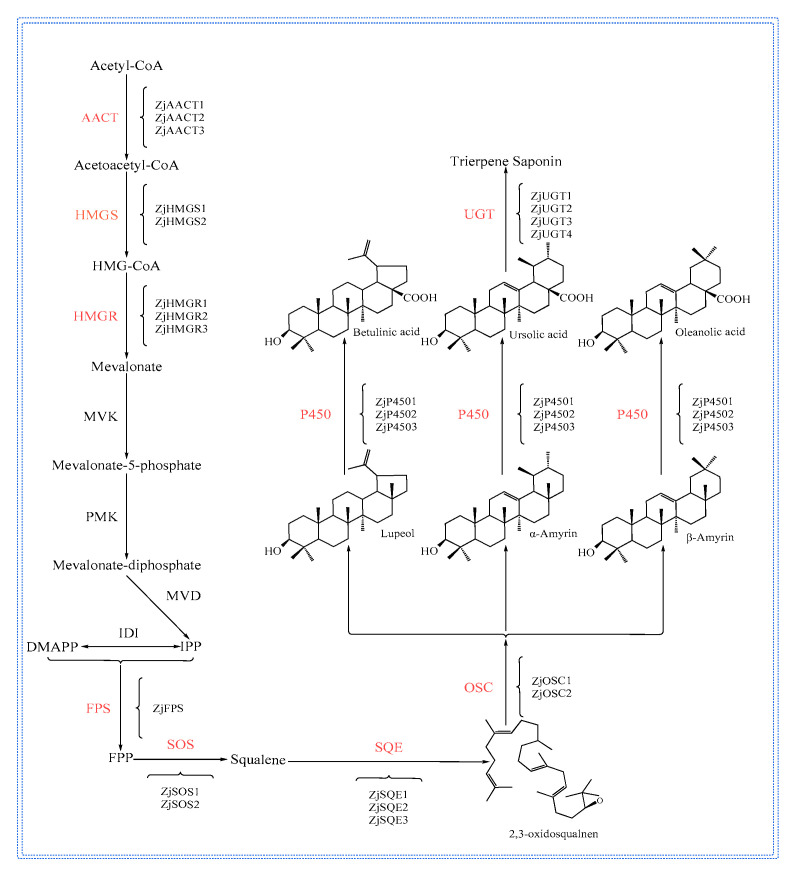
Major intermediates and key genes in the biosynthetic pathways of ZJ and ZA terpenoids.

**Table 1 plants-12-01501-t001:** Triterpenes in ZJ and ZA.

Number	Type	Compound Name	Source	Reference
**1**	Lupane	Betulinic acid	ZJ and ZA	[18,19,20]
**2**	Lupane	Alphitolic acid	ZJ and ZA	[18,19,20]
**3**	Lupane	Betulin	ZJ and ZA	[19,20]
**4**	Lupane	Betulonic acid	ZJ and ZA	[19,20,21]
**5**	Lupane	Lupeol	ZA	[23,29]
**6**	Lupane	Methyl betulinate	ZA	[23]
**7**	Lupane	Alphitolic acid methyl ester	ZA	[22,30]
**8**	Lupane	3-O-[9(Z)-octadecenoyl] betulinic acid	ZJ	[31]
**9**	Lupane	2-O-protocatechuoylalphitolic acid	ZJ and ZA	[27,32]
**10**	Lupane	2α-Hydroxypyracrenic acid	ZA	[32]
**11**	Lupane	Platanic acid	ZA	[30]
**12**	Lupane	2α,3β,20-Trihydroxylupane-28-oic acid	ZJ	[33]
**13**	Lupane	3β,30-Dihydroxylup-20(29)-en-28-oic acid	ZJ	[33]
**14**	Lupane	(3α,4β,5β,8α,9β,10α,13α,14β,15β)-13-Carboxy-4,9-dimethyl-15-(1-methylethenyl)-3-(1-methylethyl)-18-norandrostane-4-propanoic acid	ZJ	[33]
**15**	Lupane	3-O-Trans-p-coumaroyl alphitolic acid	ZJ	[34]
**16**	Lupane	3-O-Cis-p-coumaroyl alphitolic acid	ZJ	[34]
**17**	Lupane	2-O-Benzoylalphitolic acid	ZJ	[27]
**18**	Lupane	2-O-Trans-p-coumaroylalphitolic acid	ZJ	[27]
**19**	Lupane	2-O-Cis-p-coumaroylalphitolic acid	ZJ	[27]
**20**	Lupane	2-O-Vanilloylalphitolic acid	ZJ	[27]
**21**	Lupane	2-O-p-Hydroxybenzoylalphitolic acid	ZJ	[27]
**22**	Lupane	3-O-Protocatechuoylalphitolic acid	ZJ	[27]
**23**	Oleanane	Oleanolic acid	ZJ and ZA	[18,19,21]
**24**	Oleanane	Oleanonic acid	ZJ and ZA	[19,20,21]
**25**	Oleanane	Maslinic acid	ZJ and ZA	[19,24]
**26**	Oleanane	3-O-Trans-p-coumaroyl maslinic acid	ZJ	[21,33]
**27**	Oleanane	3-O-Cis-p-coumaroyl maslinic acid	ZJ	[21,34]
**28**	Oleanane	Hydroxyoleanonic acid lactone	ZA	[19]
**29**	Oleanane	11-Oxo-maslinic acid	ZJ	[33]
**30**	Oleanane	2-O-Trans-p-coumaroyl maslinic acid	ZJ	[33]
**31**	Oleanane	3,4-Seco-olean-12-ene-3,28-dioic acid	ZJ	[33]
**32**	Oleanane	2α-Cis-p-coumaroyloxy-2α,3β,23α-trihydroxy-olean-12-en-28-oic acid	ZJ	[33]
**33**	Ursane	Pomolic acid	ZJ and ZA	[19,20]
**34**	Ursane	Pomonic acid	ZJ and ZA	[19,20,24]
**35**	Ursane	Pomolic acid 28-methyl ester	ZJ	[20]
**36**	Ursane	Ursolic acid	ZJ and ZA	[19,24,35]
**37**	Ursane	Ursonic acid	ZJ and ZA	[18,19]
**38**	Ursane	Corosolic acid	ZA	[19]
**39**	Ursane	Cecropiacic acid	ZJ and ZA	[19,33]
**40**	Ursane	3β,13β-dihydroxy-urs-11-en-28-oic acid	ZA	[25]
**41**	Ursane	3β-hydroxy-urs-20(30)-en-28-oic acid	ZA	[25]
**42**	Ursane	2α,3β-dihydroxy-urs-20(30)-en-28-oic acid	ZA	[25]
**43**	Ursane	3β,12β,13β-trihydroxy-ursan-28-oic acid	ZA	[25]
**44**	Ursane	2α,3β,13β-trihydroxy-urs-11-en-28-oic acid	ZA	[25]
**45**	Ursane	2α,3β,13β,23-tetrahydroxy-urs-11-en-28-oic acid	ZA	[25]
**46**	Ursane	2α,3β,28-trihydroxy-urs-20(30)-ene	ZA	[25]
**47**	Ursane	2α,3β,12β,13β-tetrahydroxy-ursan-28-oic acid	ZA	[25]
**48**	Ursane	Euscaphic acid	ZJ	[33]
**49**	Ursane	2β,19α-Hydroxyursolic acid	ZJ	[33]
**50**	Ursane	Jacoumaric acid	ZJ	[33]
**51**	Ursane	2-Oxopomolic acid	ZJ	[33]
**52**	Ursane	(1S,2S,4aR,4bS,6aS,9R,10S,10aS,12aR)-6a-carboxy-1,2,3,4,4a,4b,5,6,6a,7,8,9,10,10a,12,12a-hexadecahydro-1,4a,4b,9,10-pentamethyl-2-(1-methylethyl)-1-chrysenepropanoic acid	ZJ	[33]
**53**	Ceanothane	Epiceanothic acid	ZJ	[18,19,30]
**54**	Ceanothane	Ceanothic acid	ZJ	[18,19,30]
**55**	Ceanothane	Ceanothic acid 2-methyl ester	ZJ	[20]
**56**	Ceanothane	Ceanothic acid 28-methyl ester	ZJ	[20]
**57**	Ceanothane	Zizyberanal acid	ZJ	[35]
**58**	Ceanothane	Zizyberanalic acid/colubrinic acid	ZJ and ZA	[34,36]
**59**	Ceanothane	Isoceanothic acid	ZJ	[29]
**60**	Ceanothane	3-O-Protocatechuoylceanothic acid	ZJ	[32]
**61**	Ceanothane	Ceanothenic acid	ZJ and ZA	[18,26]
**62**	Ceanothane	Zizyberenalic acid	ZJ	[21,34]
**63**	Ceanothane	3-O-Vanilloylceanothic acid	ZJ	[27]
**64**	Ceanothane	24-Hydroxyceanothic acid	ZJ	[27]
**65**	Ceanothane	3-Dehydroxyceanothetric acid	ZJ	[27]
**66**	Ceanothane	3-Dehydroxyceanothetric acid 2-methyl ester	ZJ	[27]
**67**	Ceanothane	Ceanothetric acid 2-methyl ester	ZJ	[27]
**68**	Ceanothane	Epiceanothic acid 2-methyl ester	ZJ	[27]
**69**	Ceanothane	3-O-Methylzizyberanalic acid	ZJ	[27]
**70**	Ceanothane	3-O-Protocatechuoylceanothic acid 2-methyl ester	ZJ	[27]
**71**	Ceanothane	3-O-Vanilloylepiceanothic acid	ZJ	[27]
**72**	Ceanothane	3-O-Vanilloylceanothic acid 2-methyl este	ZJ	[27]
**73**	Ceanothane	3-O-p-Hydroxybenzoylceanothic acid	ZJ	[27]
**74**	Ceanothane	3-O-p-Hydroxybenzoylepiceanothic acid	ZJ	[27]
**75**	Ceanothane	2-O-Protocatechuoylisoceanothanolic acid	ZJ	[27]
**76**	Ceanothane	3-Dehydroxyceanothan-27α-carboxy-28β,19β-olide	ZJ	[27]
**77**	Ceanothane	3-O-Protocatechuoylceanothan-28β,19β-olide	ZJ	[27]
**78**	Ceanothane	2,28-Dinor-24-hydroxylup-1,17(22)-dien-27-oic acid	ZJ	[27]
**79**	Ceanothane	7β-O-Vanilloyl-3-dehydroxyceanothetric acid 2-methyl ester	ZJ	[27]

**Table 2 plants-12-01501-t002:** Saponins from ZJ and ZA.

Number	Type	Compound Name	Source	Reference
**1**	I	Jujuba saponin I	ZJ and ZA	[38,39]
**2**	I	Jujuba saponin II	ZJ	[38]
**3**	I	Jujuba saponin III	ZJ	[38]
**4**	I	Ziziphin	ZJ	[38]
**5**	I	Ziziphus saponin I	ZJ and ZA	[40,41]
**6**	I	Ziziphus saponin II	ZJ and ZA	[40,41]
**7**	I	Ziziphus saponin III	ZJ	[40]
**8**	I	Acetyljujuboside B	ZA	[42]
**9**	I	Jujuboside A	ZA	[43]
**10**	I	Jujuboside B	ZA	[43]
**11**	I	Jujuboside A_2_	ZA	[44]
**12**	I	Jujuboside C	ZA	[45]
**13**	I	Jujuboside A_1_	ZA	[42,46]
**14**	I	Jujuboside B_1_	ZA	[43,46]
**15**	I	Jujuboside I	ZA	[45,46]
**16**	I	Jujuboside D	ZJ	[47]
**17**	I	Jujuboside E	ZJ	[47]
**18**	I	Jujuboside H	ZJ	[48]
**19**	I	Jujuboside G	ZJ	[48]
**20**	I	Jujuboside F	ZJ	[48]
**21**	I	Jujuboside J	ZJ	[48]
**22**	I	Christinin A	ZJ	[49]
**23**	I	Christinin C	ZJ	[49]
**24**	II	Jujuboside III	ZA	[46]
**25**	II	Jujuboside IV	ZA	[46]
**26**	II	Jujubasaponin IV	ZJ	[40]
**27**	II	Jujubasaponin V	ZJ	[40]
**28**	III	Jujubasaponin VI	ZJ and ZA	[39,40]
**29**	IV	LotosideI	ZJ	[49]
**30**	IV	Lotoside II	ZJ	[49]
**31**	IV	Lotoside III	ZJ	[49]
**32**	V	Portojujuboside A	ZA	[50]
**33**	V	Portojujuboside B	ZA	[50]
**34**	V	Portojujuboside B_1_	ZA	[50]
**35**	VI	Jujuboside H	ZA	[51]
**36**	VI	Jujuboside G	ZA	[52]
**37**	VII	Jujuboside E	ZA	[53]
**38**	VIII	Jujuboside II	ZA	[46]
**39**	IX	3-O-[(β-D-glucopyranosyl (1→2)-β-D-glucopyr-anosyl-]-20-O-[(β-D-xylopyranosyl-(1→6)-β-D-glucopyranosyl]-2α,3β,12β-trihydroxy-dammar-25-en-24-one	ZJ	[49]
**40**	X	3-O-[α-L-rhamnopyranosyl (1→2)-α-L-arabinopyranosyl-]-30 -[β-D-gluco-pyranosyl-(1→2)-β-D-glucopyranosyl]-3β,25,30-O-trihydroxy-16-one-20R,24R-epoxydammarane	ZJ	[49]
**41**	X	3-O-[α-Lrhamnopyranosyl- (1→4)-α-L-arabinopyranosyl-]-30-O-[β-D-glucopyranosyl-(1→2)-β-D-glucopyranosyl]-3β,25,30-trihydroxy-16-one-20R,24R-epoxydammarane	ZJ	[49]
**42**	XI	3-O-α-L-rhamnopyranosyl-(1→4)-α-L -rhamnopyr-anosyl-(1→2)-β-D-glucopyranosyl sidrigenin	ZJ	[49]

**Table 3 plants-12-01501-t003:** Biological activities of triterpenoids from ZJ and ZA.

Biological Activity	Active Ingredient/Substance	Classification	Source	References
Anti-cancer activity	Pomlic acid, betulinic acid, 3-O-trans-p-coumaroyl alphitolic acid, oleanolic acid, 3-O-cis-p-Coumaroyl alphitolic acid, 2α,3β,19α-Trihydroxy-urs-12-en-28-oic acid, 2α,3β-Dihydroxy-urs-20(30)-en-28-oic acid, 2α,3β,28-Trihydroxy-urs-20(30)-ene	Triterpenes	ZJ and ZA	[68,69,70,71]
Antioxidant activity	Jujuboside B, 2α,3β-Dihydroxy-urs-20(30)-en-28-oic acid, 2α,3β,28-Trihydroxy-urs-20(30)-ene	Saponins and terpenoids	ZJ and ZA	[25,77]
Anti-inflammatory activity	Jujuboside F, Jujuboside G, Jujuboside H, Jujuboside J, Jujuboside I, Jujuboside A_1_, Jujuboside A, Jujuboside B, Jujuboside C, Jujubasaponin IV, Zizyberanalic acid, ceanothic acid, Alphitolic acid, oleanolic acid 3-O-trans-Coumaroyl alphitolic acid, oleanolic acid	Saponins and terpenoids	ZJ	[49,85,86]
Anti-viral activity	Betulinic acid	Triterpenes	ZJ	[80]
Antimicrobial activity	Total saponin compounds	Saponins	ZA	[87]
Renoprotective activity	3-Dehydroxyceanothetric acid 2-methyl ester	Triterpenes	ZJ	[89]
Anti-asthmatic activity	Jujuboside B	Saponins	ZA	[90]

## Data Availability

No new data were created or analyzed in this study. Data sharing is not applicable to this article.

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
