# Peer review of "Triterpenoids in Jujube: A Review of Composition, Content Diversity, Pharmacological Effects, Synthetic Pathway, and Variation during Domestication"

_plants, 2023, doi:10.3390/plants12071501_

Round 1

Reviewer 1 Report (Previous Reviewer 2)

Dear Editors/Authors,

Please, find questions and suggestions in the file.

Kind regards

Author Response

Reviewer 2 Report (New Reviewer)

Fuxu Pan et al. discussed a review article on triterpenoids in Jujube and their pharmacological effects, including biosynthetic pathways. The authors thoroughly examined the triterpenoid chemical constituents of Jujube. The manuscript needs to be revised before acceptance for publication in Plants. Some of the comments are shown below.

1.

Title: Why does “Diver-Sity” have a hyphen? Please check it.

2.

Abstract: What are the promising active triterpenoids from Jujube? Please discuss them.

3.

Introduction: The authors should discuss the significance of triterpenoids from the medicinal chemistry perspective and how these triterpenoids play a crucial lead role in developing new drugs or clinical candidates. Please refer to the following articles. Bioorganic Chemistry 82 (2019) 306–323 (https://doi.org/10.1016/j.bioorg.2018.10.039) & EJMC 188 (2020) 111974 (https://doi.org/10.1016/j.ejmech.2019.111974),

4.

It would be good if the authors provided a table form to discuss the biological activities of triterpenoids from jujube.

5.

Keep compound numbers bold throughout the manuscript.

6.

Figures 1 & 2: some chemical structures are blurred and not transparent, and bonds are enlarged. Please revise it.

Round 2

Reviewer 1 Report (Previous Reviewer 2)

Dear Authors/Editors,

The manuscript is greatly improved and just needs some final language checks that I'm sure the team will do during the proofreading. Thank you for your effort.

This manuscript is a resubmission of an earlier submission. The following is a list of the peer review reports and author responses from that submission.

Round 1

Reviewer 1 Report

Manuscript summarized 19 the types, dynamic changes and activity experimental studies of triterpenoids of jujube to provide 20 references for future research. However, there are a number of problems with the manuscript that may cause it to be scientific and readable. The compounds shown in Figure 1 are incomplete and confusing, and this part is not the focus and takes up a lot of space. The synthesis pathway and regulation of terpenoids in jujube should be the focus of the manuscript, but it is not mentioned in the manuscript. The manuscript lacks classic papers on terpenoids, and some discussion contents are shallow, failing to put forward the author's outlook and opinions.

Reviewer 2 Report

Dear Editors,
Presented MS deals with jujube triterpenoids and their pharmacological effects.

Authors should address the following issues:

Introduction section is quite short for a review, omitting other similar works in the field;

In the Results (that probably should be renamed on Results and Discussion) some of the topics are given in much detail and others are briefly mentioned (see Antibacterial activity) that should be balanced in some way. Especially of the Authors would like to keep the title of the paper. Additionally rearangement of short sections is needed; Domestication section should be broaden with selection of both species.

Discussion is simply a short conclusion.

Technical mistake are many and should be re-checked as well.

Please, see the attached file for details.
